# Effects of brain training on brain blood flow (The Cognition and Flow Study—CogFlowS): protocol for a feasibility randomised controlled trial of cognitive training in dementia

Lucy Beishon,[1] Rachel Evley,[2] Ronney B Panerai,[1,3] Hari Subramaniam,[4] Elizabeta Mukaetova-Ladinska,[5] Thompson Robinson,[1,3] Victoria Haunton[1,3]

For numbered affiliations see end of article.

**Correspondence to**
Dr Lucy Beishon;
lb330@le.ac.uk

## ABSTRACT

**Introduction** Cognitive training is an emerging non-pharmacological treatment to improve cognitive and physical function in mild cognitive impairment (MCI) and early Alzheimer's disease (AD). Abnormal brain blood flow is a key process in the development of cognitive decline. However, no studies have explored the effects of cognitive training on brain blood flow in dementia. The primary aim of this study is to assess the feasibility for a large-scale, randomised controlled trial of cognitive training in healthy older adults (HC), MCI and early AD.

**Methods and analysis** This study will recruit 60 participants, in three subgroups of 20 (MCI, HC, AD), from primary, secondary and community services. Participants will be randomised to a 12-week computerised cognitive training programme (five × 30 min sessions per week), or waiting-list control. Participants will undergo baseline and follow-up assessments of: mood, cognition, quality of life and activities of daily living. Cerebral blood flow will be measured at rest and during task activation (pretraining and post-training) by bilateral transcranial Doppler ultrasonography, alongside heart rate (3-lead ECG), end-tidal $CO_2$ (capnography) and beat-to-beat blood pressure (Finometer). Participants will be offered to join a focus group or semistructured interview to explore barriers and facilitators to cognitive training in patients with dementia. Qualitative data will be analysed using framework analysis, and data will be integrated using mixed methods matrices.

**Ethics and dissemination** Bradford Leeds Research Ethics committee awarded a favourable opinion (18/YH/0396). Results of the study will be published in peer-reviewed journals, and presented at national and international conferences on ageing and dementia.

**Trials registration number** NCT03656107; Pre-results.

## INTRODUCTION

The incidence of dementia is rising as the population ages, with 46.8 million people living with dementia worldwide.[1 2] By 2030, this is projected to rise to 75 million.[1 2] However, there are currently few diagnostic or therapeutic strategies available to offer

---

### Strengths and limitations of this study

► This study uses a mixed methods approach to investigate the feasibility of cognitive training in healthy older adults, mild cognitive impairment and Alzheimer's disease.

► This study includes an assessment of the effects of cognitive training on neurovascular function, by transcranial Doppler ultrasonography.

► This study includes a broad range of outcome measures to take a holistic approach to evaluating cognitive training in dementia.

► The primary aim of this study is to identify the feasibility of this protocol in patients with cognitive impairment, and as such is not a fully powered trial.

► This study is limited to identifying these outcomes in healthy older adults, mild cognitive impairment and Alzheimer's disease with mild-to-moderate deficits.

---

these patients. Dementia is a progressive condition characterised by gradual loss of cognitive and/or non-cognitive higher functions (ie, language, visuospatial, attention).[3] Mild cognitive impairment (MCI) is characterised by subjective and objective decline in cognitive function, but with preserved functional independence in daily living.[4] It has become increasingly recognised that deranged vascular function is an early contributor to the deposition of amyloid plaques and tau tangles in Alzheimer's disease (AD), and that these pathologies exacerbate one another (two-hit hypothesis).[5] In a recent systematic review and meta-analysis, we demonstrated clear abnormalities in vascular function across a number of imaging modalities at the MCI stage.[6] Thus, treatments that can improve cerebral perfusion or vascular function could represent an early treatment option for dementia.[6] In this study, the target

populations are: healthy older adults (as controls), adults with AD and MCI, to capture the effects of cognitive training on vascular physiology at an early stage in the dementia process.

The Cognition and Flow Study (CogFlowS) is a feasibility, parallel, randomised controlled trial (RCT) to examine the use of a cognitive training programme in adults with MCI, early AD and healthy older adults (HC). In addition to measuring cognitive and functional outcomes, this study will also use transcranial Doppler ultrasonography (TCD) to measure changes in cerebral blood flow (CBF) before and after a 12-week cognitive training programme.

Cognitive interventions can be considered as three broad categories: cognitive training, cognitive rehabilitation and cognitive stimulation.[7 8] Cognitive training describes a structured, guided programme of standardised tasks designed to provide practice or training within a specified cognitive domain(s), to translate into functional benefits.[7 8] This is distinguished from cognitive rehabilitation, where patient-centred approaches and goals are used to improve everyday function, rather than improve cognition function,[7–9] and cognitive stimulation which aims to generally improve cognition, social function and quality of life.[10] Cognitive training is attractive in that it offers a cost-effective, non-invasive and acceptable intervention to patients, with no reported adverse effects.[11] However, high-quality RCT evidence is lacking in studies of cognitive training in dementia.[8 9 12 13] Two previous Cochrane reviews[8 9] highlighted that the evidence was sparse, with many of the studies underpowered, of low-to-moderate quality and with relatively few for MCI.[8 9] Recent systematic reviews of cognitive training in MCI have demonstrated moderate benefits in verbal learning and memory, global cognition, non-verbal learning, working memory, attention and psychosocial functioning.[12 14] Thus far, studies of cognitive training in dementia have been heterogeneous in study design, participants included, type and definition of training, outcome measures and duration and intensity of intervention.[8 9 13 15 16] This hampers the adequate meta-analysis of data, and of conclusions that can be drawn from pooled analyses.[13 17] Few studies have included neuroimaging outcomes to explore brain plasticity or neural mechanisms underlying changes in cognition after cognitive training.[17] Therefore, the CogFlowS addresses the development of a future high-quality RCT of cognitive training in dementia, and aims to fill the evidence gap on the use of TCD to assess neurovascular plasticity.

TCD is a non-invasive, ultrasound-based imaging modality that allows measurements of CBF velocity (CBFv) at rest[18] and following activation by cognitive tasks.[19] TCD has a number of advantages over other functional imaging techniques, such as: MRI and positron emission tomography (PET), including lower cost, portability, relative ease of operator training, high temporal resolution and higher acceptability to patients.[20 21] TCD is particularly suited to older people, and those with dementia, as it can be used in patients with pacemakers, metal implants and claustrophobia.[20] The use of TCD is a novel method to investigate neurovascular training effects and for markers predictive of which patients may benefit from particular cognitive training programmes. This would allow the development of a personalised approach to developing cognitive training programmes for people with dementia.

The use of neuroimaging outcomes has also been limited in cognitive training studies,[17 22] but understanding the training effects occurring at the physiological level may provide important mechanistic information to any potential benefits conferred by cognitive training.[23–25] Belleville *et al* demonstrated increased parietal activation in patients with MCI, following a multimodal cognitive training programme designed to improve episodic memory.[23] In a near-infrared spectroscopy study by Vermeij *et al*, patients with MCI had greater training benefits at high load working memory where participants had greater prefrontal activation,[26] whereas baseline global and hippocampal atrophy were predictive of poorer training gains in healthy older adults and those with MCI.[24] Therefore, imaging biomarkers can potentially be used to predict and tailor interventions to individual patient's needs.[24 25] While a number of studies have demonstrated positive structural and functional effects of cognitive training,[25 27 28] they have largely focused on healthy older adults,[27] with the use of structural[27] or functional MRI[23 24] and PET;[29 30] there have been no TCD studies examining changes in neurovascular response to cognitive training in patients with cognitive impairment. In addition to the inclusion of neuroimaging, this study uses a mixed methods approach. This study includes a qualitative component to explore any benefits, problems or barriers to engaging with the training programme, and to identify any benefits not measured by traditional outcomes.

The primary objective for this study is to determine the feasibility for a large-scale RCT of cognitive training in HC, AD and MCI. Secondary objectives seek to identify any clinical benefits of a cognitive training programme in terms of: activities of daily living, cognition, mood, quality of life and TCD-measured task activation responses in brain blood flow. In addition, the qualitative study seeks to explore barriers and facilitators to the cognitive training programme, how programmes can be adapted to support participation for patients living with dementia and any additional benefits not measured through traditional methods as perceived by patients and their carers. Finally, we plan to explore the lived experience of the patient and their carer, and the impact cognitive training has on them and their life.

## METHODS AND ANALYSIS

This protocol was reported in accordance with the Standard Protocol Items: Recommendations for Interventional Trials checklist,[31] which is included in online supplementary information.

## Patient and public involvement

During the design phase of this study, a patient and public involvement (PPI) group was consulted, comprised research network volunteers from the Alzheimer's society who were either people living with cognitive impairment or their carers. The group reviewed the study design, research questions and outcome measures, and a number of changes were made as a result. These included: providing an option of a focus group or interview to participants, changes to the recruitment procedures and the design of the cognitive training programme. All study participants will receive a lay summary at the end of the study explaining what the outcomes of the study were. A further PPI meeting was held following successful funding to refine the study documentation and procedures for participants. The trial will be overseen by a trial steering committee (TSC), consisting of patient and public members, and independent academics. The TSC will review the trial conduct on a six monthly basis. As this is a feasibility trial, the burden of intervention is to be assessed in terms of its acceptability to patients and their carers.

## Sample selection

HC will be recruited through poster advertisement, friends, family and carers of enrolled patient participants, Join Dementia Research and the research interested list at the University of Leicester. Participants with a diagnosis of MCI, or AD, will be recruited through poster advertisement, secondary care services (outpatient clinics within the University Hospitals of Leicester (acute) and the Leicestershire Partnership (mental health) National Health Service Trusts), research interested lists, Join Dementia Research and local community groups. In addition, participants will be sought through letter invitation from their general practitioner (GP) surgery. Participants will be allowed a minimum of 24 hours to decide if they would like to enrol in the study, and to contact the researcher or research delivery team at the Leicestershire Partnership Trust, who will arrange for them to undergo an eligibility assessment, formal consent and study enrolment. As part of the eligibility assessment, participants will undergo a screening cognitive assessment (Montreal Cognitive Assessment—MoCA) to ensure deficits are classified as mild to moderate. Participants will be consented for this, and the MoCA will not form part of the baseline assessments. For participants who lack capacity to consent to the study, a personal consultee declaration will be completed by a friend, relative or carer, in addition to verbal assent from the participant themselves where possible. Flow charts of the recruitment process and participant procedures can be seen in figures 1 and 2, respectively.

The planned recruitment target for this study is 60 participants, divided into three subgroups of 20 (HC, AD

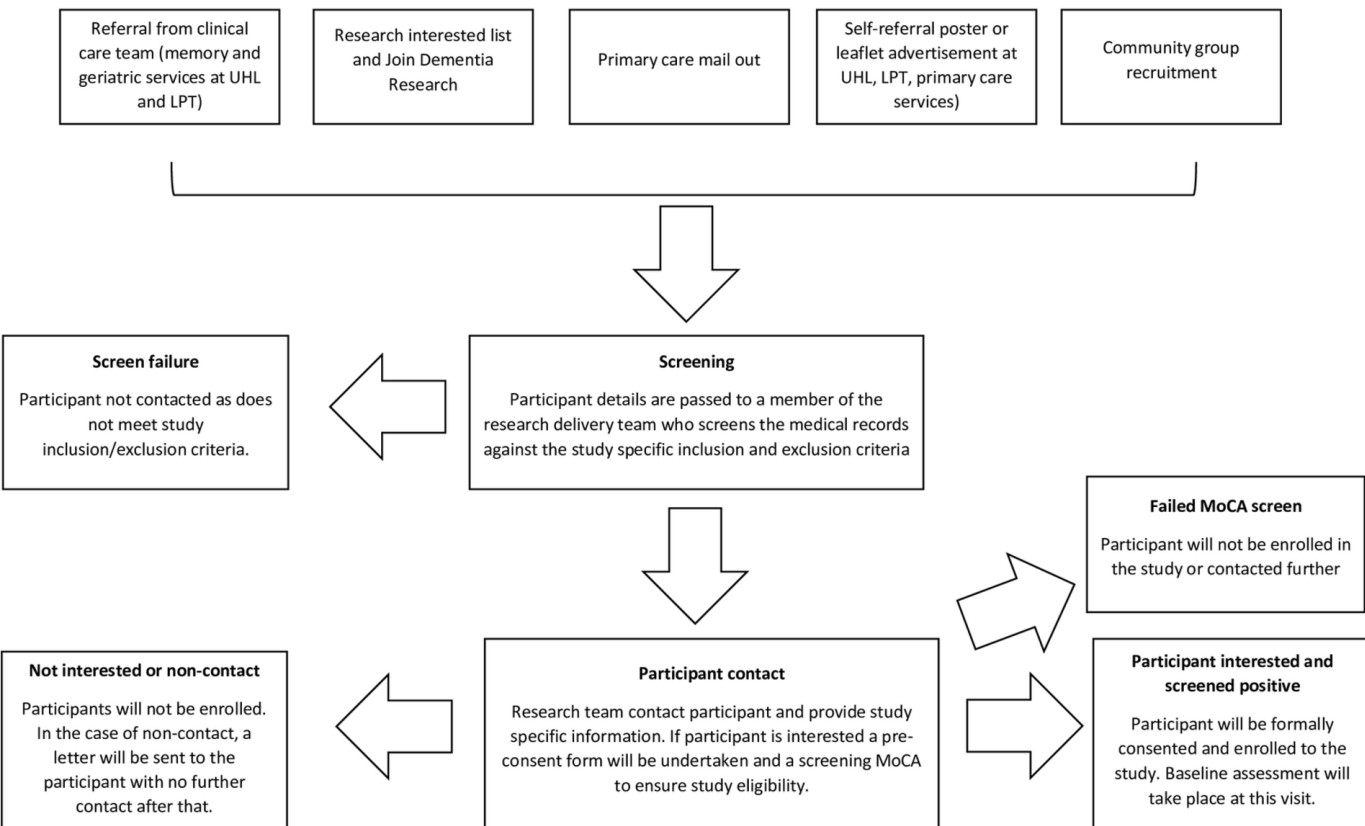

**Figure 1** Flow chart of study recruitment procedures. LPT, Leicestershire Partnership Trust; MoCA, Montreal Cognitive Assessment; UHL, University Hospitals of Leicester.

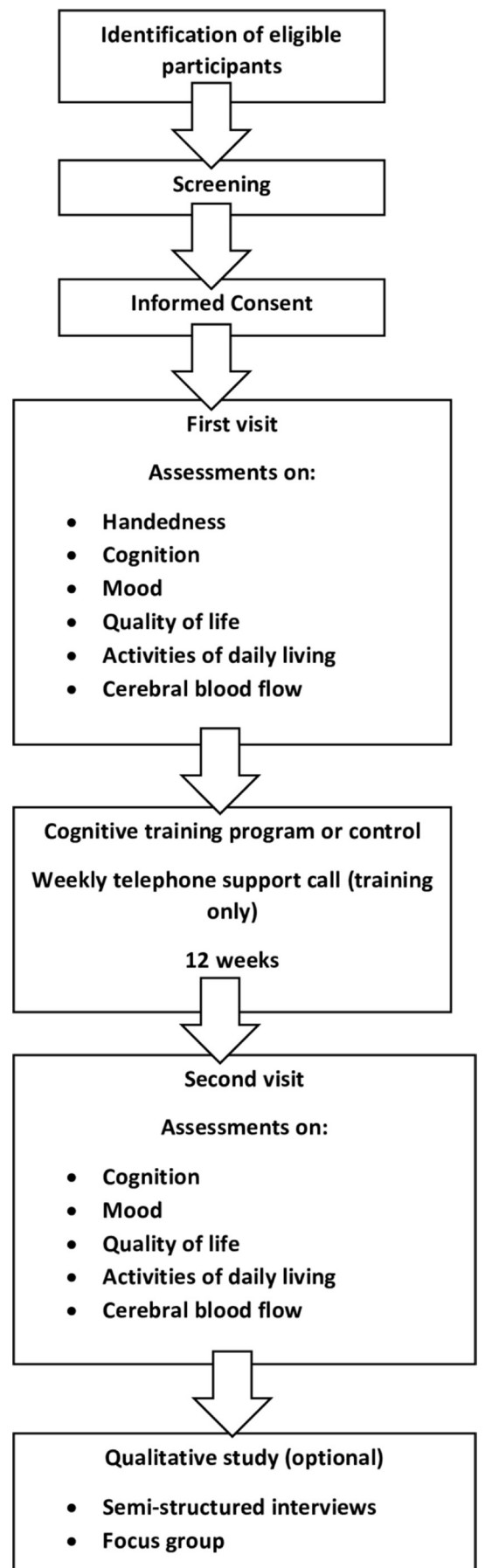

**Figure 2** Flow chart of participant procedures during the study.

and MCI). As the primary aim of this study is feasibility, a formal sample size calculation has not been performed. Recruiting between 24 and 50 participants for a feasibility study is acceptable and therefore the planned target of 60 participants is sufficiently large enough while giving a margin for drop-outs and loss to follow-up.[32 33]

The following inclusion and exclusion criteria were designed to identify a sample of participants that will be able to engage with a cognitive training programme, and to exclude any major medical comorbidities that can affect CBF:

### Inclusion criteria
1. Healthy controls will be free of any medical comorbidity or medication that could adversely affect cognition. Volunteers with well-controlled comorbidities (ie, hypertension, diabetes, will be considered for inclusion).
2. MCI as defined by NIA/AA 2011[34] and Petersen criteria.[4]
3. AD as defined by the NIA/AA 2011 criteria.[34]

And:
4. Deficits will be mild to moderate as defined by MoCA score of >9 for participants with MCI and AD.[35–37]
5. Willing to participate.
6. Capacity to consent to the study/personal consultee.
7. Patients on and off antidementia medications will be included (acetylcholinesterase inhibitors, glutamate receptor antagonists).
8. Good understanding of written and spoken English.
9. Age>50 years.
10. Access to the internet and a computer/laptop device.

### Exclusion criteria
1. Healthy controls with any medical comorbidity or medication that could adversely affect cognition, or poorly controlled medical comorbidities (ie, hypertension, diabetes).
2. Unwilling to take part.
3. Unable to consent/no personal consultee.
4. Major medical comorbidity: severe heart failure (ejection fraction <20%), carotid artery stenosis, severe respiratory disease, major stroke.
5. Pregnancy, planning pregnancy or lactating.
6. Inadequate bilateral TCD windows.
7. Participants already enrolled into other interventional studies that would confound study results
8. Insufficient understanding of written and spoken English.
9. Age<50 years.
10. No access to the internet and a computer/laptop device.

### Intervention
Participants randomised to the intervention will complete a 12-week cognitive training programme. Participants will be required to complete five, 30 min sessions per week for 12 weeks. The cognitive training programme will be provided by Lumosity as part of a collaboration through

the Human Cognition Project. Lumosity is commercially available software, developed by a group of neuropsychologists, which has been used across several studies of brain training and disciplines.[38–40] These studies have been included in a number of systematic reviews and meta-analysis.[12 41] It is a multidomain, online-based cognitive training tool, which is relatively easy to use and administer. It has been designed to adapt to the individual's cognitive function in order to administer the training programme to their individual level of cognitive function. Exercises will be selected from Lumosity's commercially available online software which have been preclassified to target the following cognitive domains: attention, memory, visuospatial, verbal fluency and language. These five cognitive domains will be evaluated through formal cognitive testing, and through the neurovascular coupling assessment. Compliance will be monitored through Lumosity online software which will log and track the number of minutes and times per week a participant has spent training, in order to calculate the dose of training to which each participant has been exposed.

### Randomisation

Randomisation will be performed using Sealed Envelope, by the researcher. This is an online-based randomisation tool which uses random permuted blocks to allocate participants to waiting list control or intervention. Participants will be enrolled and assigned a patient identiication number (PIN) consecutively, and randomised to a treatment arm corresponding to the PIN. Given the nature of the trial, it is not possible to blind participants to the intervention. The investigator will be providing weekly telephone support for the intervention group, in addition to undertaking baseline and follow-up measurements, and therefore blinding of the investigator is also not possible. However, data analysis will be blinded by generating a batch-anonymised dataset. Randomisation will be undertaken at the initial visit. Once participants have been randomised to the intervention arm, they will be provided with information on how to access and use the cognitive training programme at home.

### Data collection and integration

All participants (intervention and control groups) will be required to attend two assessments: baseline and follow-up at 12-weeks post training. Assessments will be carried out at the participant's home, or designated research space at the Leicestershire Partnership Trust, or Cerebral Haemodynamics in Ageing and Stroke Medicine (CHiASM) research space. Participants will be given the option to divide the assessments across two visits if necessary. This is to reduce any burden of participation for people living with dementia, who may fatigue more quickly with the baseline and follow-up assessments. It is anticipated that the majority of participants will be able to complete all assessments in one visit, and this will be included within the feasibility assessment for the study.

Data will be collected on demographics, medical history and medication use. Assessments of cognition (Addenbrooke's Cognitive Examination III (ACE-III)), mood (Geriatric Depression Scale 15), activities of daily living (Lawton instrumental activities of daily living) and quality of life (Dementia Rated Quality of Life) will be carried out at both visits. An assessment of neurovascular function will be carried out at the CHiASM research space, a temperature-controlled room, free of noise and distraction. Participants will be seated throughout the protocol. All measurements will be made at rest (5 min baseline recording), and during cerebral activation (selected cognitive tasks). CBFv will be measured continuously by insonation of the bilateral middle cerebral arteries (MCA) (DWL Doppler Box), using 2 MHz probes, fixed in place with a headframe. Beat-to-beat blood pressure (BP) will be measured continuously using a Finometer cuff on the non-dominant middle finger (Finapres Medical Systems, Amsterdam, The Netherlands), and calibrated using a brachial BP recording (UA767 BP monitor). End-tidal $CO_2$ ($ETCO_2$) will be measured by capnography using nasal cannulae (Capnochek Plus), and the R-R interval will be recorded using a 3-lead ECG. The task activation protocol has been published previously.[42 43] In brief, participants will be presented with five tasks selected from the ACE-III, and percentage increase in CBFv from resting baseline will be calculated at task initiation. The tasks will be selected for five cognitive domains: attention, memory, visuospatial, language and verbal fluency, based on recent studies conducted by this group.[43 44] Data will be collected on the following parameters: average CBFv, $ETCO_2$, HR and BP at rest over 5 min, and peak percentage change from baseline in all parameters at 5–10 s, and 10–20 s after task activation for each of the five cognitive tasks.

At least 1 week after completion of the follow-up assessment, participants with a diagnosis of MCI, or AD, will be invited to attend a semistructured interview or focus group. The schedule of questions can be seen in online supplementary information. The schedule of questions has been framed around the six constructs of the health belief model (risk susceptibility, risk severity, benefits to action, barriers to action, self-efficacy and cues to action)[45] to evaluate the cognitive training programme, and the experience of the participant and their carer. Interviews and focus groups will be recorded continuously with a digital audio recorder, and notes of non-verbal and paralinguistic clues will be made. All digital recordings will be transcribed within 5–7 days of the interview or focus group. Semistructured interviews are an iterative process, following the first couple of interviews the transcripts will be analysed and the themes and concepts emerging from these initial data will be further explored at the following interviews.

Data security and storage will be conducted in line with general data protection regulations (https://eugdpr.org/), and details regarding data security and storage can be seen in online supplementary information.

## Trial discontinuation

This is not a drug intervention trial, and therefore interim analysis will not be required. Criteria for trial discontinuation are as follows: ineligibility, significant protocol deviation, significant non-adherence to the programme, an adverse event or disease progression resulting in inability to comply with study procedures, consent withdrawn or lost to follow-up.

## Data analysis and interpretation

The difference for each participant between baseline and follow-up assessment for each of the continuous outcome measures described above will be reported along with the overall mean difference.

CBFv data will be recorded and stored in the PHYSIDAS system, and analysed offline. Data will be visually inspected for quality, and files rejected where quality is deemed to be poor, with reasons. Peak percentage change in CBFv will be calculated relative to a 20 s baseline, prior to task initiation.

The findings from the semistructured interviews and focus groups will be evaluated using framework analysis.[41] The digital recordings of the interviews and focus groups will be transcribed verbatim, the transcripts will be read in detail, that is, line by line, and open codes will be formed categorising and conceptualising the responses and identifying the major themes. Two researchers LB and RE will independently code the first few transcripts to ensure consistency in coding.

Following this initial coding, the analytical framework will be developed; this is an iterative process and will develop through coding of additional transcripts. Once the final transcript has been coded, the analytic framework will be used to generate the framework matrix. The framework matrix will be developed in NVivo V.11 (QSR International), and allow for the recognition of patterns and outliers within the data. Respondent validation will be used to establish a degree of correspondence between the researcher's views and those of the research participant.[46]

Data from the quantitative and qualitative arms of the study will be integrated using mixed methods approaches of triangulation and mixed methods matrices.[47]

## Statistical analysis and sample size calculation

Data will be checked for normality, and non-normally distributed data will be appropriately transformed, and parametric tests applied. Data will be reported as mean (SD) for continuous variables, and number (percentage) for nominal data. Differences in baseline demographics between the intervention and control groups will be analysed within population group (HC/AD/MCI) using $X^2$ for nominal data, and independent t-testing for continuous data. Any baseline differences in demographics between population groups will also be reported. To assess the impact of the intervention and population group on each of these differences, a two-way analysis of variance will be carried out. The interaction between population group and treatment group will be analysed to assess if the difference between the treatment groups is different between the population groups. Post-hoc testing will be conducted by Tukey to analyse for main effects. Data will be recorded in Microsoft Excel. Statistical analyses will be performed using the latest version of SPSS for Windows (SPSS V.24), and graphs will be produced using the latest version of GraphPad Prism for Windows.

## Study limitations

This is a feasibility study, and thus will not be powered adequately to detect changes in the secondary end points outlined above. However, given the novel aspects of this study protocol, such as the inclusion of a TCD-measured changes in CBF responses, and the mixed methods design, this requires evaluation of the feasibility before moving to a larger, fully powered trial of cognitive training in dementia. Furthermore, this study will include assessments of patient experience and tolerability to facilitate this. Although TCD affords excellent temporal resolution in the measurement of CBFv, the MCA supplies blood to approximately 80% of the cerebral cortex, and therefore reflects a global measure of perfusion, and cannot discriminate where changes are occurring, due to poor spatial resolution. Techniques such as MRI, and PET scans afford better spatial resolution, but have their own limitations, particularly in older patients with cognitive deficits. TCD-measured CBFv relies on the assumption that the vessel diameter remains constant, despite fluctuations in $CO_2$ and BP.[18] This study elected to focus on healthy older adults, MCI and AD in the first instance. Deficits in CBFv have been demonstrated in both MCI and AD, but inclusion of participants with vascular cognitive impairment would provide further information on the capacity for plasticity to neurovascular physiology among dementia subtypes. In addition, only participants with mild-to-moderate deficits will be included, given that data thus far do not support the use of cognitive training in more advanced dementia.[9 12] Only participants who are undertaking the training programme will be offered telephone support, as it is primarily to support technical issues with delivery of the programme. However, this could introduce a placebo effect, and future studies would ideally correct for this by providing support to both groups. Finally, one investigator is undertaking both baseline and follow-up assessments, and providing telephone support, which carries a risk of introducing researcher bias into the study. As this is primarily a feasibility study, this is of lesser importance here, but a future trial should ideally have separate team members conducting these different roles to minimise the risk of bias.

## ETHICS AND DISSEMINATION
## Safety considerations

All participants with capacity in this study will provide informed, written consent, and the study will be conducted in accordance with the Declaration of Helsinki and Good Medical Practice guidelines.[48] Given the nature of

dementia, not all participants will have capacity to consent to the study. In this instance, a personal consultee (friend, relative or carer) will be consulted for their opinion on whether the participant is likely to have wanted to participate in this study, and they will be asked to sign a consultee declaration form. It was not considered ethical to exclude participants on the grounds of capacity, given the nature of the disease studied.

## Dissemination

The results from this study will be submitted to peer-reviewed journals in ageing and dementia research, and presented at national and international conferences on these themes. We anticipate that the results will be presented and published in late 2020, early 2021. In addition, the results will be disseminated to members of the community through the Alzheimer's Society, community groups, podcasts and social media.

## Trial committees and monitoring

The study will be monitored at six monthly intervals throughout its duration by a TSC, independent of the funder, consisting of lay and independent academic members. Each site will have a lead investigator who is responsible for the identification and recruitment of participants, data collection and completion for CRFs, follow-up measurements and adherence to study protocols. Serious adverse events (SAEs) will be reported to the sponsor within one working day of identification, and the sponsor will report all serious unexpected or adverse reactions to the ethics committee concerned. The chief investigator will submit an annual report to the ethics committee which lists all SAEs. Where necessary, protocol amendments will be submitted to the study sponsor or ethics committee for approval.

**Author affiliations**
¹Cardiovascular Sciences, University of Leicester College of Medicine Biological Sciences and Psychology, Leicester, UK
²Division of Clinical Neuroscience, University of Nottingham, Nottingham, UK
³NIHR Leicester Biomedical Research Centre, British Heart Foundation Cardiovascular Research Centre, University of Leicester, Leicester, UK
⁴The Evington Centre, Leicestershire Partnership NHSTrust, Leicester, UK
⁵Department of Neuroscience, Psychology and Behaviour, University of Leicester, Leicester, UK

**Acknowledgements** The authors would like to thank the members of the patient and public involvement group from the Alzheimer's Society for their contribution to the development of this protocol.

**Contributors** The quantitative arm of the study was conceived and designed by LB, VH, RBP and TR. LB and RE designed the qualitative arm of the study. HS and EM-L reviewed and edited the protocol. All authors assisted in drafting this manuscript.

**Funding** This study was reviewed and funded by the Dunhill Medical Trust, and LB is a Dunhill research training fellow (RTF1806\27). TR is an NIHR Senior Investigator. This funding source had and will not have a role in the design, conduct, analyses, interpretation of the data or decision to submit results. The cognitive training programme is being provided by Lumosity free of charge, as part of an industry-institution collaboration through the Human Cognition Project. Lumosity is providing advice and support on the composition and nature of the programme. However, this study is financially independent of Lumosity, and Lumosity cannot influence or change the results or outcomes of this study.

**Competing interests** None declared.

**Patient consent for publication** Not required.

**Ethics approval** Ethical approval was obtained from the Bradford Leeds Research Ethics Committee (18/YH/0396).

**Provenance and peer review** Not commissioned; externally peer reviewed.

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
