## [Reviewer comments · BMJ Open]

ARTICLE DETAILS

TITLE (PROVISIONAL)	The Effects of Brain Training on Brain Blood Flow (The Cognition and Flow Study -CogFlowS): A Protocol for A Feasibility Randomised Controlled Trial of Cognitive Training in Dementia.
AUTHORS	Beishon, Lucy; Evley, Rachel; Panerai, Ronney; Subramaniam, Hari; Mukaetova-Ladinska, Elizabeta; Robinson, Thompson; Haunton, Victoria

VERSION 1 – REVIEW

REVIEWER	Jen Mozolic Warren Wilson College, USA
REVIEW RETURNED	12-Dec-2018

GENERAL COMMENTS	Blinding - in this feasibility study, neither participants nor researchers will be blinded, only data analysis will be performed blindly. Are there any creative ways to limit placebo effects and minimize the difference in experiences of the experimental and control groups? Will control group receive weekly support phone calls and some placebo activity? Also, will you monitor control group to ensure that they do not increase cognitive activity or begin training independently concurrent with the start of the trial? Trial Registration - ...will be completed within six ?months? of recruitment of the first study participant
--

REVIEWER	Mitchell McMaster Australian National University
REVIEW RETURNED	27-Feb-2019

GENERAL COMMENTS	I'd like to commend the authors on a very well written protocol and well thought-out novel study. I have no major changes but some small changes to help clarify the study for the reader and some queries/comments for the authors to consider. Being a feasibility study many of the changes suggested are less important for this study, but it is often useful to run pilots and feasibility studies as close to the protocol of the upcoming, full study to assess feasibility most accurately. P5, L13- Did you mean "and"? cognitive and non-cognitive P5, L47- The authors have listed a training duration of 8-12 weeks? Will the training program differ in length for participants? Or is a final decision on the duration not yet made? Please clarify and justify if the training program will be of variable duration for participants. P5, L50-52 You provide a good summary sentence of cognitive training. Can you please provide a one sentence on each of
---

	cognitive rehabilitation and cognitive stimulation, just for completeness and differentiation? P5, L55- Did you mean “within”? P6, 37- A bracket was opened but not closed. Using “such as” may be better than brackets? P8, L6- Typo: SPIRIT P8, L40- Please provide numerical scores for mild-moderate deficits. 9-26? P8, L50- 20 (as numerals), use abbreviation “HC”, already introduced. P9, L41- Including anti-dementia medications is less of a problem for the feasibility study (where outcomes are less important), but this may represent a confound in the larger study and authors may wish to consider this for the larger study. P9, L51- Considering that the intervention is based on computerised brain training, having sufficient computer skills may be an important inclusion criteria to add to the access to computer/internet. P9- Exclusion criteria: Perhaps excluding participants who have used computerised brain training in the last 6 months? P10, L8- Mixed dementia (Alzheimer’s disease with elements of vascular dementia) is quite common. Will participants with known mixed dementia be excluded? P10, L34- 5 x 30 minute sessions/week is a quite a lot more training than most studies. Even with less training than this compliance/dosage is often a complication. With this level of training I would anticipate problems with compliance especially with the impaired populations due to time constraints/fatigue. This would potentially result in the lowest rates of compliance in more impaired participants which would be problematic for interpretation of results. The amount of testing will be investigated in the qualitative component, it might be best to ask participants how much testing would be acceptable if you proceed with 5 x 30 minutes. P10, L49- Could you please elaborate on how exercises will be selected? You also said that the training program would be personalised to the participant. By this do you mean that different participants will be doing different tasks due to how they perform in different domains? If so, without a very large sample size this would result in very “dilute” effects of the intervention. These effects are generally small, when detected. I think focussing on certain domains for all participants may yield better results (more important for the full study). P11, L19- The same investigator should not provide telephone support to the intervention group, as well as do follow-up assessments. This introduces a definite bias. Is it possible that different team members carry out these separate roles? Could you please elaborate on the support that will be provided? Will this be only for participants who are experiencing problems or will all participants receive phone calls? P11, L32- Shouldn’t the control group attend 3 testing sessions? Eg Wk 1. Intervention/control: baseline testing; Wk 8/12. Intervention: follow up testing, control: pre-training testing; wk 16/24. Control: follow-up testing. It might be helpful for the reader to include a diagram showing, intervention periods, testing periods, qualitative component and weeks that all of these elements occur for intervention & control? P11, L36 & P11, L50- L36 you say that there are two assessment sessions (baseline and follow-up) that can occur at CHIASM offices or the participants’ home, but then L50 you say there will be
--	---

	neurovascular assessments at the CHiASm offices. Do these assessments occur at different times? Why would all testing not be on one occasion at CHiASM offices? P12,L12- Will ACE-III will be an outcome measure and a stimulus for CBFv changes? or will CBFv measurements occur while baseline cognitive data is being collected? Please clarify. If ACE-III will serve as both, but cognitive data will be collected separately to CBFv data, I would suggest a different cognitive tests to avoid practice effects, which would likely occur differentially between the three groups and introduce bias. P12,L22-The qualitative component includes questions about the testing periods. Please change “cognitive training program” to read “follow-up testing” P12, L25- Will you offer the participants the opportunity to participate in an interview or a focus group? I would suggest that interviews would be the best option. The participants who are most likely to experience the most problems due to cognitive impairment, are also the least able to contribute in a focus group situation. For these participants interviews with carers and PwD would probably be the best approach. It is also probably best to collect data in the same manner for all participants ie interviews. P14, L20- Given that you will be running a two-way ANOVA to look at the effects of intervention and population, are the three t-tests necessary? The ANOVA will determine the presence of effects and post-hoc analyses will be able to identify the individual comparisons which are significant. P15, L39- Please include one sentence about verbal assent also being obtained for people who do not possess the capacity to provide their own consent, in addition to the written consent of the personal consultee.
--	---

VERSION 1 – AUTHOR RESPONSE

Reviewer 1

I'd like to commend the authors on a very well written protocol and well thought-out novel study. I have no major changes but some small changes to help clarify the study for the reader and some queries/comments for the authors to consider. Being a feasibility study many of the changes suggested are less important for this study, but it is often useful to run pilots and feasibility studies as close to the protocol of the upcoming, full study to assess feasibility most accurately.

Comment 1

P5, L13- Did you mean “and”? cognitive and non-cognitive

Response 1

Thank you, we have amended this to ‘and/or’ to reflect that patients can present with or without non-cognitive deficits.

Comment 2

P5, L47- The authors have listed a training duration of 8-12 weeks? Will the training program differ in length for participants? Or is a final decision on the duration not yet made? Please clarify and justify if the training program will be of variable duration for participants.

Response 2

The duration of the programme was under development with Lumosity© at the time of submission and is now fixed at a 12-week duration. This duration was selected based on their previous research of sufficient “dose” of training for benefit. This has been amended throughout the protocol.

Comment 3

P5, L50-52 You provide a good summary sentence of cognitive training. Can you please provide a one sentence on each of cognitive rehabilitation and cognitive stimulation, just for completeness and differentiation?

Response 3

Thank you, this information has been added for completeness.

Comment 4

P5, L55- Did you mean “within”?

Response 4

Thank you this has been corrected to within.

Comment 5

P6, 37- A bracket was opened but not closed. Using “such as” may be better than brackets?

Response 5

This has been changed from brackets to ‘such as’.

Comment 6

P8, L6- Typo: SPIRIT

Response 6

Thank you, this has been amended.

Comment 7

P8, L40- Please provide numerical scores for mild-moderate deficits. 9-26?

Response 7

An amendment has been requested to have only a lower cut-off score. The original aim for including cut-offs was to prevent patients with very severe cognitive function being inappropriately enrolled into the study. However, from a practical perspective many patients do not neatly fit into these cut offs as a result of varying degrees of pre-morbid cognitive function. In addition, the distinction between MCI and Alzheimer's disease is based on functional status, which is a clinical, not cognitive, assessment. Therefore, in order to improve the accuracy of diagnosis for enrolled participants we will enrol participants based on their clinical diagnosis, rather than cut-off scores, and these have been amended in the protocol.

Comment 8

P8, L50- 20 (as numerals), use abbreviation "HC", already introduced.

Response 9

This has been amended accordingly.

Comment 10

P9, L41- Including anti-dementia medications is less of a problem for the feasibility study (where outcomes are less important), but this may represent a confound in the larger study and authors may wish to consider this for the larger study.

Response 10

Thank you for the comment. We agree that this would be an important confounder and would be a potential concern in a larger, fully powered trial. We did not feel it was ethical to stop these medications for a feasibility study, and practically speaking the majority of patients with a diagnosis of Alzheimer's disease will be established on these medications.

Comment 11

P9, L51- Considering that the intervention is based on computerised brain training, having sufficient computer skills may be an important inclusion criteria to add to the access to computer/internet.

Response 11

We are only enrolling participants with adequate technology available as listed in the inclusion/exclusion criteria, and as such, they will most likely be familiar with their own technology. Participants are provided with a demonstration of the cognitive training prior to starting and a user/troubleshooting guide. The programme itself requires minimal computer skills to be able to access and use it, and has been designed in a simple manner to avoid issues around lack of computer skills. In addition, a useful part of the feasibility assessment would be identifying this as a potential issue and how common it is.

Comment 12

P9- Exclusion criteria: Perhaps excluding participants who have used computerised brain training in the last 6 months?

Response 12

As the primary aim of this study is feasibility, and the trial is not powered to determine changes in effect sizes, this is of a lesser concern in this instance. However, this is an important point and something we would certainly consider incorporating into a larger, fully-powered trial.

Comment 13

P10, L8- Mixed dementia (Alzheimer's disease with elements of vascular dementia) is quite common. Will participants with known mixed dementia be excluded?

Response 13

At this stage we are only enrolling participants with Alzheimer's disease, and so participants with mixed dementia will be excluded. In future studies it would be beneficial to include this patient group, in addition to vascular dementia. The criteria quoted (NIA/AA 2011) will cover this as an exclusion.

Comment 14

P10, L34- 5 x 30 minute sessions/week is a quite a lot more training than most studies. Even with less training than this compliance/dosage is often a complication. With this level of training I would anticipate problems with compliance especially with the impaired populations due to time constraints/fatigue. This would potentially result in the lowest rates of compliance in more impaired participants which would be problematic for interpretation of results. The amount of testing will be investigated in the qualitative component, it might be best to ask participants how much testing would be acceptable if you proceed with 5 x 30 minutes.

Comment 15

Thank you. We agree that this is a significant dosage. This was taken on the recommendation from Lumosity© concerning their previous dose effect research data. This is in part, one of the reasons for conducting a feasibility trial to investigate if this dosage is feasible and at which levels of impairment (i.e. healthy, MCI, or established dementia). As part of the trial we will be measuring compliance (conducted through the CT software). In addition, this information will be beneficial going forward to identify the most appropriate or benefiting group for cognitive training. Thank you for the suggestion, we will certainly include the compliance and dose under feasibility and barriers to training in the qualitative component of the study.

Comment 16

P10, L49- Could you please elaborate on how exercises will be selected? You also said that the training program would be personalised to the participant. By this do you mean that different participants will be doing different tasks due to how they perform in different domains? If so, without a very large sample size this would result in very "dilute" effects of the intervention. These effects are generally small, when detected. I think focussing on certain domains for all participants may yield better results (more important for the full study).

Response 16

The exercises are selected from Lumosity's© commercially available software. The exercises have been classified by Lumosity© under specific cognitive domains that they are intended to target and train. Thus, we are targeting five specific cognitive domains with the task-activation protocol and have selected tasks from Lumosity's© software which have been categorised under these domains. All participants will train on the same exercises, but at different levels depending on their cognitive ability.

This is a design feature of the software so that it adapts to each participant. The information in this section has been updated to clarify this further.

Comment 17

P11, L19- The same investigator should not provide telephone support to the intervention group, as well as do follow-up assessments. This introduces a definite bias. Is it possible that different team members carry out these separate roles?

Response 17

Unfortunately, for this study we are resource limited to one investigator completing the assessments. This is in part due to the skills needed to undertake the neurovascular coupling protocol. As the primary outcome is feasibility, we feel this is an important concern and one which has now been highlighted in the limitation section. As a feasibility study, the bias is of a lesser concern, but certainly if this study is taken forward separate team members would be needed to complete assessments and provide support.

Comment 18

Could you please elaborate on the support that will be provided? Will this be only for participants who are experiencing problems or will all participants receive phone calls?

Response 18

The telephone support is mainly to troubleshoot issues occurring with technology as this is a primary concern for the feasibility study. Participants are being offered a weekly call, however, it is optional and so some participants may decline this if they feel it is too intrusive. As the controls are not undertaking any intervention we are not providing this service to them. We understand that ideally this would be the case to match the controls more closely to the intervention, however resources do not permit this number of weekly phone calls to be undertaken. This has been added to the limitations section.

Comment 19

P11, L32- Shouldn't the control group attend 3 testing sessions?
Eg Wk 1. Intervention/control: baseline testing; Wk 8/12. Intervention: follow up testing, control: pre-training testing; wk 16/24. Control: follow-up testing. It might be helpful for the reader to include a diagram showing, intervention periods, testing periods, qualitative component and weeks that all of these elements occur for intervention & control?

Response 19

Due to time constraints, a cross over trial was not possible so the control group will only be attending two sessions. Only participants who have completed the training programme will be invited for interview after the follow-up assessment is complete. A flow chart has been provided in Figure 2.

Comment 20

P11, L36 & P11, L50- L36 you say that there are two assessment sessions (baseline and follow-up) that can occur at CHiASM offices or the participants' home, but then L50 you say there will be neurovascular assessments at the CHiASM offices. Do these assessments occur at different times?

Why would all testing not be on one occasion at CHiASM offices?

Response 20

Ideally, all assessments will be completed at the same time at the CHiASM office. However, for participants with dementia who may fatigue more quickly, we have included the option for them to split the assessments across 2 visits if necessary. This is a common practice for dementia research studies within our locality. However, we anticipate the majority of participants will be able to complete all the assessments in one sitting. Again, this will be part of the feasibility assessment to understand level of burden the assessments place on participants with cognitive impairment.

Comment 21

P12,L12- Will ACE-III will be an outcome measure and a stimulus for CBFv changes? or will CBFv measurements occur while baseline cognitive data is being collected? Please clarify. If ACE-III will serve as both, but cognitive data will be collected separately to CBFv data, I would suggest a different cognitive tests to avoid practice effects, which would likely occur differentially between the three groups and introduce bias.

Response 21

The cognitive test scores will be collected at the same time as the CBFv measurements, thus cognitive tasks will not be repeated and will be compiled with the remaining ACE-III tasks to generate a full cognitive assessment. Therefore, there are no concerns for practice effects.

Comment 22

P12,L22-The qualitative component includes questions about the testing periods. Please change "cognitive training program" to read "follow-up testing"

Response 22

This has been changed accordingly.

Comment 23

P12, L25- Will you offer the participants the opportunity to participate in an interview or a focus group? I would suggest that interviews would be the best option. The participants who are most likely to experience the most problems due to cognitive impairment, are also the least able to contribute in a focus group situation. For these participants interviews with carers and PwD would probably be the best approach. It is also probably best to collect data in the same manner for all participants ie interviews.

Response 23

We agree that interviews would be most appropriate for patients with cognitive impairment and their carers. The focus group option was a recommendation taken on board by the PPI group to provide choice to patients with different preferences. We plan to offer the option of both to participants and identify which is the patient preferred option to take forward in future studies. We anticipate this will most likely be interviews, as suggested by the reviewer, given that they can be conducted more flexibly in the participants' home.

Comment 24

P14, L20- Given that you will be running a two-way ANOVA to look at the effects of intervention and population, are the three t-tests necessary? The ANOVA will determine the presence of effects and post-hoc analyses will be able to identify the individual comparisons which are significant.

Response 24

Thank you, the wording in this section is confusing and the t-tests are for differences in baseline demographics, and the differences in outcome measures are by ANOVA, with post-hoc tests. This section has been updated to clarify this.

Comment 25

P15, L39- Please include one sentence about verbal assent also being obtained for people who do not possess the capacity to provide their own consent, in addition to the written consent of the personal consultee.

Response 25

A statement to this effect has been added under sample selection.

Editorial requests:

Comment 1

- Along with your revised manuscript, please provide an English language examples of the patient consent form as a supplementary file as per item #32 of the SPIRIT checklist.

Response 1

This has been included as a supplementary file.

Reviewer(s)' Comments to Author:

Reviewer: 1

Reviewer Name: Jen Mozolic

Institution and Country: Warren Wilson College, USA

Please state any competing interests or state 'None declared': none declared

Please leave your comments for the authors below

Comment 1

Blinding - in this feasibility study, neither participants nor researchers will be blinded, only data analysis will be performed blindly. Are there any creative ways to limit placebo effects and minimize the difference in experiences of the experimental and control groups? Will control group receive weekly support phone calls and some placebo activity? Also, will you monitor control group to ensure

that they do not increase cognitive activity or begin training independently concurrent with the start of the trial?

Response 1

We agree that lack of blinding is a significant limitation to the study, and is a common problem for cognitive training studies where an adequate active control condition is still under debate. Unfortunately for this study we are resource limited to one investigator completing the assessments. This is in part due to the skills needed to undertake the neurovascular coupling protocol. As the primary outcome is feasibility, we feel this is an important concern and one which has now been highlighted in the limitation section. As a feasibility study, the bias is of a lesser concern, but certainly if this study is taken forward separate team members would be needed to complete assessments and provide support. The telephone support is mainly to troubleshoot issues occurring with technology as this is a primary concern for the feasibility study. Participants are being offered a weekly call, however, it is optional and so some participants may decline this if they feel it is too intrusive. As the controls are not undertaking any intervention we are not providing this service to them. We understand that ideally this would be the case to match the controls more closely to the intervention, however resources do not permit this number of weekly phone calls to be undertaken. This has been added to the limitations section.

Contamination from the control group is a risk with studies of cognitive training, however, as the control group are receiving the intervention for free (waiting listed) they are incentivised to remain in the study as such and not to obtain the training through other means (where they would invariably have to pay to access).

Comment 2

Trial Registration - ...will be completed within six ?months? of recruitment of the first study participant

Response 2

Thank you this has been amended to months.

Reviewer: 2

Reviewer Name: Mitchell McMaster

Institution and Country: RCT Manager, Centre for Research on Ageing, Health and Wellbeing, NHMRC Centre for Research Excellence in Cognitive Health, Australian National University, Australia.

Please state any competing interests or state 'None declared': None

Please leave your comments for the authors below Please see attached word document for comments

FORMATTING AMENDMENTS (if any)

Required amendments will be listed here; please include these changes in your revised version:

Comment 1

1. Patient and Public Involvement:

We have implemented an additional requirement to all articles to include 'Patient and Public Involvement' statement within the main text of your main document. Please refer below for more information regarding this new instruction:

Authors must include a statement in the methods section of the manuscript under the sub-heading 'Patient and Public Involvement'.

This should provide a brief response to the following questions:

How was the development of the research question and outcome measures informed by patients' priorities, experience, and preferences?

How did you involve patients in the design of this study?

Were patients involved in the recruitment to and conduct of the study?

How will the results be disseminated to study participants?

For randomised controlled trials, was the burden of the intervention assessed by patients themselves?

Patient advisers should also be thanked in the contributorship statement/acknowledgements. If patients and or public were not involved please state this.

Response 1

This section has now been added to the methods in line with the guidance provided above and we have thanked the patient contributors in the acknowledgements section.

Comment 2

2. Please re-upload your supplementary files in PDF format.

Response 2

These have now been uploaded in pdf format.

Comment 3

3. Please remove all your figures in your main document and upload each of them separately under file designation 'Image' (except tables and please ensure that Figures are of better quality or not pixelated when zoom in). NOTE: They can be in TIFF, JPG or PDF format and make sure that they have a resolution of at least 300 dpi. Figures in DOCUMENT, EXCEL and POWER POINT format are not acceptable.

Response 3

The figures have been converted to PDF format separate from the original manuscript.

VERSION 2 – REVIEW

REVIEWER	Mitchell McMaster Australian National University
REVIEW RETURNED	26-Mar-2019

GENERAL COMMENTS	I'd like to thank the authors for their responses to my questions. They have clarified several aspects of the paper and I think it reads a lot more clearly. I look forward to the results of the full study in the future. I think there is only one paragraph that still requires some minor clarifications:  -In the data collection paragraph could you please include a few words to indicate that CBFv measures will be obtained while participants undergo the cognitive testing? -Could you please include some of the information you have included in response 20? Eg assessments will be carried out at CHiASM offices but if participants become fatigued there is the option to complete this testing later at home. There were two remaining instances where "8" needs to be removed from "8-12 weeks": "Data collection and Intergration" paragraph and "Methods and Analysis" section of abstract.
---

VERSION 2 – AUTHOR RESPONSE

Reviewer: 2

Reviewer Name: Mitchell McMaster

Institution and Country: Centre for Research on Ageing, Health and Wellbeing, NHMRC Centre for Research Excellence in Cognitive Health, Australian National University, Australia.

Please state any competing interests or state 'None declared': None declared

Please leave your comments for the authors below I'd like to thank the authors for their responses to my questions. They have clarified several aspects of the paper and I think it reads a lot more clearly. I look forward to the results of the full study in the future.

I think there is only one paragraph that still requires some minor clarifications:

Comment 1

-In the data collection paragraph could you please include a few words to indicate that CBFv measures will be obtained while participants undergo the cognitive testing?

Response 1

Thank you for the comment. We have added the following statement to clarify the exact measures that will be obtained: Data will be collected on the following parameters: average CBFv, ETCO₂, HR, and BP at rest over five minutes, and peak percentage change from baseline in all parameters at 5-10 seconds, and 10-20 seconds after task activation for each of the five cognitive tasks.

Comment 2

-Could you please include some of the information you have included in response 20? Eg assessments will be carried out at CHiASM offices but if participants become fatigued there is the option to complete this testing later at home.

Response 2

Participants will be given the option to divide the assessments across two visits if necessary. This is to reduce any burden of participation for people living with dementia, who may fatigue more quickly with the baseline and follow-up assessments. It is anticipated that the majority of participants will be able to complete all assessments in one visit, and this will be included within the feasibility assessment for the study.

Comment 3

There were two remaining instances where "8" needs to be removed from "8-12 weeks": "Data collection and Intergration" paragraph and "Methods and Analysis" section of abstract.

Response 3

Thank you, this has been amended accordingly.